# Detecting critical treatment effect bias in small subgroups

**Piersilvio de Bartolomeis** [1]   **Javier Abad** [1]   **Konstantin Donhauser** [1]   **Fanny Yang** [1]

## Abstract

Randomized trials are considered the gold standard for making informed decisions in medicine, yet they often lack generalizability to the patient populations in clinical practice. Observational studies, on the other hand, cover a broader patient population but are prone to various biases. Thus, before using an observational study for decision-making, it is crucial to *benchmark* its treatment effect estimates against those derived from a randomized trial. We propose a novel strategy to benchmark observational studies beyond the average treatment effect. First, we design a statistical test for the null hypothesis that the treatment effects estimated from the two studies, conditioned on a set of relevant features, differ up to some tolerance. We then estimate an asymptotically valid lower bound on the maximum bias strength for any subgroup in the observational study. Finally, we validate our benchmarking strategy in a real-world setting and show that it leads to conclusions that align with established medical knowledge.

## 1. Introduction

Randomized trials have traditionally been the gold standard for informed decision-making in medicine, as they allow for unbiased estimation of treatment effects under mild assumptions. However, there is often a significant discrepancy between the patients observed in clinical practice and those enrolled in randomized trials, limiting the generalizability of the trial results (Rothwell, 2005; Duma et al., 2018). To address this issue, the U.S. Food and Drug Administration advocates for using observational data, as it is usually more representative of the patient population in clinical practice (Platt et al., 2018; Klonoff, 2020). Yet, a major caveat to this recommendation is that several sources of bias, including hidden confounding, can compromise the

[1]Department of Computer Science, ETH Zurich. Correspondence to: Piersilvio de Bartolomeis <piersilvio.debartolomeis@inf.ethz.ch>.

*Accepted at the 1st Machine Learning for Life and Material Sciences Workshop at ICML 2024.* Copyright 2024 by the author(s).

causal conclusions drawn from observational data.

In light of the inherent limitations of randomized and observational data, it has become a popular strategy to *benchmark* observational studies against existing randomized trials to assess their quality (Dahabreh et al., 2020; Forbes & Dahabreh, 2020). The main idea behind this approach is first to emulate the procedures adopted in the randomized trial within the observational study; see e.g. (Hernán & Robins, 2016) for a detailed explanation. Then, the treatment effect estimates from the observational data are compared with those from the randomized data. If the estimates are similar, we may be willing to trust the observational study for patient populations where the randomized data is insufficient.

To support the benchmarking framework, several works propose statistical tests that compare treatment effect estimates between randomized and observational data (Viele et al., 2014; Hussain et al., 2023; De Bartolomeis et al., 2024; Yang et al., 2023; Demirel et al., 2024). In particular, two properties have been identified as essential for effective benchmarking of observational studies: *tolerance* and *granularity*. Tolerance allows the acceptance of studies with negligible bias that does not impact decision-making, thereby significantly reducing false rejections in real-world settings where some bias is expected. Granularity, on the other hand, allows the detection of bias on small subgroups or individuals that would otherwise go unnoticed.

In this work, we design a statistical test for the null hypothesis that treatment effects differ up to some tolerance value when conditioned on a relevant subset of features. Our test is the first, to our knowledge, to satisfy granularity and tolerance. Further, we use our test to estimate an asymptotically valid lower bound on the maximum bias strength for any individual. Finally, we show that our lower bound leads to conclusions that align with established medical knowledge.

## 2. Problem setting

We have access to two datasets: $D_{\mathrm{rct}}$ of size $n_{\mathrm{rct}}$ from a randomized trial (rct) and $D_{\mathrm{os}}$ of size $n_{\mathrm{os}}$ from an observational study (os), containing tuples $Z := (X, Y, T)$ of covariates $X \in \mathbb{R}^d$, bounded observed outcome $Y \in \mathbb{R}$, and treatment assignment variable $T \in \{0, 1\}$. We assume that the data is drawn i.i.d from the distributions $\mathbb{P}^{\mathrm{rct}}$ and $\mathbb{P}^{\mathrm{os}}$

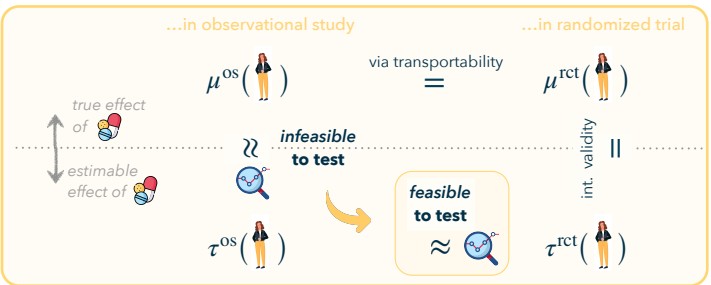

*Figure 1.* High-level illustration of our approach. We want to test if the bias in the observational study, i.e. $\mu^{\mathrm{os}} - \tau^{\mathrm{os}}$, is contained within a tolerance range. However, the true treatment effect $\mu^{\mathrm{os}}$ is not identifiable, and instead, we test the bias between the treatment effects estimated from the two studies, i.e. $\tau^{\mathrm{os}} - \tau^{\mathrm{rct}}$.

that are marginal distributions of the respective full distribution $\mathbb{P}^{\diamond}_{\mathrm{full}}$ over $(X, U, Y(0), Y(1), Y, T)$ for $\diamond \in \{\mathrm{rct}, \mathrm{os}\}$. In particular, the full distribution also includes randomness over a vector of unobserved covariates $U \in \mathbb{R}^k$ and potential outcomes $(Y(0), Y(1)) \in \mathbb{R}^2$. We further assume that the support of the randomized trial is included in the support of the observational study, i.e. $\mathrm{supp}(\mathbb{P}^{\mathrm{rct}}_X) \subseteq \mathrm{supp}(\mathbb{P}^{\mathrm{os}}_X)$.

**Treatment effect estimation**   A crucial quantity to estimate for decision-making in many domains is the conditional average treatment effect (CATE):

$$\mu^{\diamond}(x) := \mathbb{E}_{\mathbb{P}^{\diamond}_{\mathrm{full}}} \left[ Y(1) - Y(0) \mid X = x \right],$$

for $\diamond \in \{\mathrm{rct}, \mathrm{os}\}$ and $\mathcal{X} \subseteq \mathrm{supp}\left(\mathbb{P}^{\mathrm{rct}}_X\right)$. Unfortunately, we cannot estimate the CATE from the observed data as we never observe the potential outcomes. Instead, we can estimate the regression function defined by

$$\tau^{\diamond}(x) = \mathbb{E}_{\mathbb{P}^{\diamond}} \left[ Y \mid T = 1, X = x \right] - \mathbb{E}_{\mathbb{P}^{\diamond}} \left[ Y \mid T = 0, X = x \right],$$

for $\diamond \in \{\mathrm{rct}, \mathrm{os}\}$. For the treatment effect in the randomized trial, we observe that $\tau^{\mathrm{rct}}(x) = \mu^{\mathrm{rct}}(x)$ holds for all $x \in \mathcal{X}$, under the assumption of internal validity outlined below.

**Assumption 2.1** (Internal validity). *The data-generating process of the randomized trial satisfies*

$$(i) \ \ Y = Y(T) \ \ \mathbb{P}^{\mathrm{rct}}_{\mathrm{full}} - \text{almost surely.}$$
$$(ii) \ \ T \perp\!\!\!\perp (Y(1), Y(0)).$$
$$(iii) \ \ \mathbb{P}^{\mathrm{rct}}_{\mathrm{full}}(T = 1 \mid X, U) = \pi \in (0, 1).$$

In particular, Assumption 2.1 is expected to hold in a completely randomized experiment, and thus, $\mu^{\mathrm{rct}}$ can be estimated from the observed data under mild assumptions (Rubin, 1978). On the other hand, we cannot estimate $\mu^{\mathrm{os}}$ from the observed data due to hidden confounding or other sources of bias in the observational study, i.e. we cannot rule out the existence of $x \in \mathcal{X}$ such that $\tau^{\mathrm{os}}(x) \neq \mu^{\mathrm{os}}(x)$. Therefore, it is crucial to benchmark the observational study before using the estimate of $\tau^{\mathrm{os}}$ for any downstream task.

### 2.1. Null hypothesis

Our goal is to test if the bias in the observational study, defined as $\delta^{\star}(x) := \tau^{\mathrm{os}}(x) - \mu^{\mathrm{os}}(x)$ for all $x \in \mathcal{X}$, is contained within a tolerance range. However, the bias $\delta^{\star}$ is not estimable from the data. Instead, we can test the bias $\tilde{\delta}(x) := \tau^{\mathrm{os}}(x) - \tau^{\mathrm{rct}}(x)$, which is equivalent to $\delta^{\star}$ under internal validity and transportability, i.e. $\mu^{\mathrm{os}}(x) = \mu^{\mathrm{rct}}(x)$ for all $x \in \mathcal{X}$ (see Figure 1). We would like to test if the bias $\tilde{\delta}$ between the two studies is contained within a tolerance range (requires tolerance) across all patient subgroups (requires granularity). Hence, we will now introduce a null hypothesis that allows for both tolerance and granularity.

To do so, we define two bounded tolerance functions $\tau^{\mathrm{os}}_{\pm} : \mathcal{X} \to \mathbb{R}$ that capture how much the estimated treatment effects can differ between studies and satisfy $\tau^{\mathrm{os}}_{-}(x) \leq \tau^{\mathrm{os}}(x) \leq \tau^{\mathrm{os}}_{+}(x)$ for all $x \in \mathcal{X}$. Further, we define the patient subgroups via a subset of features $X^{\mathcal{J}}$, corresponding to the covariates with indices $\mathcal{J} \subseteq \{1, \cdots, d\}$. We can then introduce our null hypothesis, given by

$$H_0 : \ \ \mathbb{E}_{\mathbb{P}^{\mathrm{rct}}} \left[ \tau^{\mathrm{rct}}(X) \mid X^{\mathcal{J}} \right] \in \tag{1}$$
$$\left[ \mathbb{E}_{\mathbb{P}^{\mathrm{rct}}}[\tau^{\mathrm{os}}_{-}(X) \mid X^{\mathcal{J}}], \mathbb{E}_{\mathbb{P}^{\mathrm{rct}}}[\tau^{\mathrm{os}}_{+}(X) \mid X^{\mathcal{J}}] \right], \ \ \mathbb{P}^{\mathrm{rct}}_{X^{\mathcal{J}}} - \mathrm{a.s.}$$

**Discussion of our null hypothesis**   We provide several remarks on the null hypothesis in Equation (1). First, we satisfy tolerance by testing if $\tau^{\mathrm{rct}}(x)$ is contained (in probability) in an interval around $\tau^{\mathrm{os}}(x)$, for all $x \in \mathcal{X}$. Second, we can satisfy granularity by choosing an appropriate subset $\mathcal{J}$: When $|\mathcal{J}| = d$, we detect bias at the individual level, thereby satisfying the strictest definition of granularity. On the other hand, when $|\mathcal{J}| = 0$, we test if the average treatment effects are equal, thus potentially ignoring bias in small subgroups and individuals. Third, we test if the treatment effects are equal (up to tolerance) on the support of the randomized trial since we cannot extrapolate outside the support of $\mathbb{P}^{\mathrm{rct}}_X$ without further assumptions.

**Example: User-specified tolerance**   A natural choice for the tolerance functions is to add (respectively subtract) a

user-specified function $\delta(x) \geq 0$, that is

$$\tau_\pm^{\text{os}}(x) = \tau^{\text{os}}(x) \pm \delta(x), \quad \text{for all } x \in \mathcal{X}.$$

The function $\delta$ can incorporate all sources of bias in the observational study, such as unobserved confounding and non-adherence to treatment assignments. For instance, we can test whether $\|\tilde{\delta}\|_{L^\infty(\mathbb{P}_X^{\text{rct}})}$ is larger than a critical value $\delta_{\text{CT}} \in \mathbb{R}$ by choosing $\tau_\pm^{\text{os}}(x) = \tau^{\text{os}}(x) \pm \delta_{\text{CT}}$.

## 3. Methodology

In this section, we rewrite the null hypothesis from Equation (1) in terms of a *signal* function that captures the bias between $\tau^{\text{os}}$ and $\tau^{\text{rct}}$. Then, we propose an oracle test statistic assuming that the tolerance functions $\tau_\pm^{\text{os}}$ are known. Finally, we provide asymptotic guarantees for the finite-sample test statistic where the tolerance functions are estimated.

### 3.1. Null hypothesis using signal function

We first observe that, for some tolerance functions $\tau_\pm^{\text{os}}$, Equation (1) is equivalent to stating that there exists a function $g : \mathbb{R}^{|\mathcal{J}|} \to [0, 1]$ such that $\tau_g^{\text{os}}(X) := g\left(X^{\mathcal{J}}\right) \tau_+^{\text{os}}(X) + \left(1 - g\left(X^{\mathcal{J}}\right)\right) \tau_-^{\text{os}}(X)$ satisfies

$$\mathbb{E}_{\mathbb{P}^{\text{rct}}}\left[\tau^{\text{rct}}(X) \mid X^{\mathcal{J}}\right] = \mathbb{E}_{\mathbb{P}^{\text{rct}}}[\tau_g^{\text{os}}(X) \mid X^{\mathcal{J}}], \quad \mathbb{P}_{X^{\mathcal{J}}}^{\text{rct}} - \text{a.s.}$$

We test a slightly more restrictive hypothesis by assuming that $g$ lies in a sufficiently rich function class $\mathcal{G}$:

$$H_0^{\mathcal{G}} : \mathbb{E}_{\mathbb{P}^{\text{rct}}}\left[\tau^{\text{rct}}(X) \mid X^{\mathcal{J}}\right] = \mathbb{E}_{\mathbb{P}^{\text{rct}}}\left[\tau_{g^\star}^{\text{os}}(X) \mid X^{\mathcal{J}}\right],$$
$$\text{for some } g^\star \in \mathcal{G}, \quad \mathbb{P}_{X^{\mathcal{J}}}^{\text{rct}} - \text{a.s.}$$

In practice, one can either restrict $\mathcal{G}$ to a particular function class if domain knowledge is available or use neural networks as general function approximations.

We can then rewrite the null hypothesis above using a *signal* function that captures the bias between the estimates from observational and randomized data. Recall that $Z = (X, Y, T)$ is the vector of observed variables, we define

$$\psi_g(Z) = Y\left(\frac{T}{\pi} - \frac{1-T}{1-\pi}\right) - \tau_g^{\text{os}}(X)$$

and finally arrive at the null hypothesis

$$H_0^{\mathcal{G}} : \mathbb{E}_{\mathbb{P}^{\text{rct}}}\left[\psi_{g^\star}(Z) \mid X^{\mathcal{J}}\right] = 0, \tag{2}$$
$$\text{for some } g^\star \in \mathcal{G}, \quad \mathbb{P}_{X^{\mathcal{J}}}^{\text{rct}} - \text{a.s.}$$

At first glance, testing the null hypothesis in Equation (2) may seem equivalent to testing equality of conditional means (Delgado, 1993; Neumeyer & Dette, 2003; Racine et al., 2006; Luedtke et al., 2019; Muandet et al., 2020); however, we remark that this equivalence holds only if the function $g^\star$ is known, and to our knowledge, the scenario where $g^\star$ is unknown has not been previously explored.

### 3.2. Oracle test statistic

We now derive a kernelized test statistic for the null hypothesis in Equation (2). First, we observe that the hypothesis $H_0^{\mathcal{G}}$ implies an infinite set of unconditional moment constraints, i.e. for any $g \in \mathcal{G}$, it holds that

$$\mathbb{E}_{\mathbb{P}^{\text{rct}}}\left[\psi_g(Z) \mid X^{\mathcal{J}}\right] = 0, \quad \mathbb{P}_{X^{\mathcal{J}}}^{\text{rct}} - \text{a.s.} \implies$$
$$\mathbb{E}_{\mathbb{P}^{\text{rct}}}\left[\psi_g(Z)f(X^{\mathcal{J}})\right] = 0, \quad \text{for all measurable } f.$$

Therefore, the validity of testing the RHS would carry over to the validity of testing $H_0^{\mathcal{G}}$. However, testing the RHS of the implication above for all measurable functions is infeasible. Instead, we can restrict $f$ to be in a reproducing kernel Hilbert space (RKHS). The problem then becomes more tractable since it holds that

$$\mathbb{H}^2(\psi_g) := \left(\sup_{\|f\|_{\mathcal{F}} \leq 1} \mathbb{E}_{\mathbb{P}^{\text{rct}}}\left[\psi_g(Z)f(X^{\mathcal{J}})\right]\right)^2 \tag{3}$$
$$= \left\|\mathbb{E}_{\mathbb{P}^{\text{rct}}}\left[\psi_g(Z)k(X^{\mathcal{J}}, \cdot)\right]\right\|_{\mathcal{F}}^2$$
$$= \mathbb{E}_{\mathbb{P}^{\text{rct}}}\left[\psi_g(Z)k(X^{\mathcal{J}}, \tilde{X}^{\mathcal{J}})\psi_g(\tilde{Z})\right],$$

where $k$ is a uniformly bounded reproducing kernel corresponding to an RKHS $\mathcal{F}$, and $\tilde{Z}$ is an independent copy of $Z$ following the same distribution. In particular, the null hypothesis $H_0^{\mathcal{G}}$ implies that $\mathbb{H}^2(\psi_g) = 0$ for $g = g^\star$, and thus we can construct a valid test based on $\mathbb{H}^2(\psi_{g^\star})$.

**A valid test statistic** Given i.i.d. samples $Z_i$ from $\mathbb{P}^{\text{rct}}$, an unbiased empirical estimate of $\mathbb{H}^2(\psi_g)$ is the cross U-statistic (Kim & Ramdas, 2024), defined as

$$\hat{\mathbb{H}}^2(\psi_g) := \frac{2}{n_{\text{rct}}} \sum_{i=1}^{n_{\text{rct}}/2} h(Z_i; \psi_g), \quad \text{with}$$

$$h(Z_i; \psi_g) := \frac{2}{n_{\text{rct}}} \sum_{j=n_{\text{rct}}/2+1}^{n_{\text{rct}}} \psi_g(Z_i)k(X_i^{\mathcal{J}}, X_j^{\mathcal{J}})\psi_g(Z_j),$$

for all $g \in \mathcal{G}$. The main advantage of the cross U-statistic is that, for $g = g^\star$, it is asymptotically normal under the null hypothesis $H_0^{\mathcal{G}}$ and weak regularity assumptions (see Theorem 3.1), i.e. as $n_{\text{rct}} \to \infty$ it holds that

$$\sqrt{\frac{n_{\text{rct}}}{2}} \frac{\hat{\mathbb{H}}^2(\psi_{g^\star})}{\hat{\sigma}\left(\hat{\mathbb{H}}^2(\psi_{g^\star})\right)} \to \mathcal{N}(0, 1),$$

where $\hat{\sigma}\left(\hat{\mathbb{H}}^2(\psi_{g^\star})\right)$ is the finite sample estimate of the variance term defined as

$$\sigma^2\left(\hat{\mathbb{H}}^2(\psi_g)\right) := \mathbb{E}_{\mathbb{P}^{\text{rct}}}\left[\left(h(Z; \psi_g) - \mathbb{E}_{\mathbb{P}^{\text{rct}}}[h(Z; \psi_g)]\right)^2\right],$$

for all $g \in \mathcal{G}$. Observe that under the assumption that $g^\star \in \mathcal{G}$, we have

$$
\mathbb{H}^2_{\text{OPT}} := \min_{g \in \mathcal{G}} \left| \sqrt{\frac{n_{\text{rct}}}{2}} \frac{\hat{\mathbb{H}}^2(\psi_g)}{\hat{\sigma}\left(\hat{\mathbb{H}}^2(\psi_g)\right)} \right| \leq \left| \sqrt{\frac{n_{\text{rct}}}{2}} \frac{\hat{\mathbb{H}}^2(\psi_{g^\star})}{\hat{\sigma}\left(\hat{\mathbb{H}}^2(\psi_{g^\star})\right)} \right| \tag{4}
$$

Therefore, we can achieve validity by comparing $\mathbb{H}^2_{\text{OPT}}$ with the quantiles of the half-normal distribution.

### 3.3. Theoretical guarantees

Since, in practice, we do not have access to the signal function $\psi_g$, we define the finite-sample analogous as

$$
\hat{\psi}_g(Z) = Y\left(\frac{T}{\pi} - \frac{1-T}{1-\pi}\right) - \hat{\tau}^{\text{os}}_g(X), \quad \text{where}
$$

$$
\hat{\tau}^{\text{os}}_g(X) := g(X^{\mathcal{J}})\hat{\tau}^{\text{os}}_+(X) + \left(1 - g\left(X^{\mathcal{J}}\right)\right)\hat{\tau}^{\text{os}}_-(X),
$$

and $\hat{\tau}^{\text{os}}_\pm$ is a consistent estimate of $\tau^{\text{os}}_\pm$ that uses only the observational data $D_{\text{os}}$. We can then define our finite-sample test statistic as

$$
\hat{\mathbb{H}}^2_{\text{OPT}} := \min_{g \in \mathcal{G}} \left| \sqrt{\frac{n_{\text{rct}}}{2}} \frac{\hat{\mathbb{H}}^2(\hat{\psi}_g)}{\hat{\sigma}\left(\hat{\mathbb{H}}^2(\hat{\psi}_g)\right)} \right|,
$$

and the testing function $\hat{\phi}(\alpha) := \mathbb{I}\left\{\hat{\mathbb{H}}^2_{\text{OPT}} \geq z_{1-\alpha}\right\}$, where $z_\alpha$ is the $\alpha$-quantile of the half-normal distribution. Below, we provide sufficient conditions for $\hat{\phi}$ to be an asymptotically valid test.

**Theorem 3.1** (Validity of the test). *We make the following assumptions:*

*(i) The variance term is non-zero, i.e.* $\mathbb{E}_{\mathbb{P}^{\text{rct}}}\left[\psi^2_{g^\star}(Z)\,k^2(X^{\mathcal{J}}, \tilde{X}^{\mathcal{J}})\,\psi^2_{g^\star}(\tilde{Z})\right] > 0.$

*(ii) $\hat{\tau}^{\text{os}}_\pm$ satisfy $\|\tau^{\text{os}}_\pm - \hat{\tau}^{\text{os}}_\pm\|_{L^2(\mathbb{P}^{\text{rct}})} = O_{\mathbb{P}^{\text{os}}}\left(\frac{1}{\sqrt{n_{\text{os}}}}\right)$, and it holds that $\lim_{n_{\text{rct}}, n_{\text{os}} \to \infty} n_{\text{rct}}/n_{\text{os}} = 0.$*

*Then, we have that*

$$
\sqrt{\frac{n_{\text{rct}}}{2}} \frac{\hat{\mathbb{H}}^2(\hat{\psi}_{g^\star})}{\hat{\sigma}\left(\hat{\mathbb{H}}^2(\hat{\psi}_{g^\star})\right)} \to \mathcal{N}(0,1), \quad \text{as } n_{\text{rct}}, n_{\text{os}} \to \infty.
$$

*Hence, $\hat{\phi}(\alpha)$ is a valid asymptotic test at level $\alpha$ for the null hypothesis $H^{\mathcal{G}}_0$ from Equation (2).*

**Power of the test** While Theorem 3.1 only shows asymptotic validity, we further present guarantees for the asymptotic power of the test in Appendix A.2. In particular, in Theorem A.1, we show that under the alternative hypothesis

$$
H^{\mathcal{G}}_A : \inf_{g \in \mathcal{G}} \sup_{\|f\|_{\mathcal{F}} \leq 1} \mathbb{E}_{\mathbb{P}^{\text{rct}}}\left[\psi_g(Z)f(X^{\mathcal{J}})\right] > 0,
$$

the test statistic $\hat{\mathbb{H}}^2_{\text{OPT}}$ in Equation (4) grows at the typical rate of order $\sqrt{n_{\text{rct}}}$ for a fixed function class $\mathcal{G}$.

### 3.4. Benchmarking the observational study

Given the theoretical results in this section, we can now introduce our strategy to benchmark observational studies. More concretely, we choose as tolerance functions $\tau^{\text{os}}_\pm(X) = \tau^{\text{os}}(X) \pm \delta$, for some constant $\delta \in \mathbb{R}^+$, and we define a data-dependent lower bound on the bias as

$$
\hat{\delta}_{\text{LB}} := \inf_\delta\{\delta : \hat{\phi}(\alpha) = 0\}, \tag{5}
$$

where $\hat{\phi}$ depends implicitly on $\delta$ via the tolerance functions and we fix $\mathcal{J} = \{1, \ldots, d\}$. Then, under the assumptions in Theorem 3.1, it holds that

$$
\mathbb{P}\left(\tilde{\delta} \geq \hat{\delta}_{\text{LB}}\right) \geq 1 - \alpha + o_{\mathbb{P}}(1).
$$

Crucially, to benchmark the observational study, we propose to compare the lower bound on the bias against a critical value, e.g. the minimum bias strength that would explain away the estimated treatment effect in a subgroup of interest. If the lower bound is greater than the critical value, we discard the conclusions drawn from the observational study. In Section 5, we demonstrate that our strategy yields conclusions consistent with epidemiological knowledge using real-world data from the Women's Health Initiative.

## 4. Semi-synthetic experiments

### 4.1. Experimental setting

**Dataset** We evaluate our testing procedure on a semi-synthetic dataset derived from a real-world randomized trial: Hillstrom's MineThatData Email dataset (Hillstrom, 2008). By default, we use 80% of the full dataset as the os and the remaining 20% as the rct.

**Bias model** We consider three different models for the bias between studies, given by $\delta^\star(x) = \mu^{\text{os}}(x) - \tau^{\text{os}}(x)$, for all $x \in \mathcal{X}$. In Scenario 1, we consider a single subgroup with a constant bias of $\delta^\star = 60$, while the rest of os remains unbiased. In Scenario 2 (Figure 5a), we add biases of varying magnitudes across 12 subgroups where the largest bias is $\delta^\star = 60$, and it affects only 12% of the observational dataset. Finally, in Scenario 3 (Figure 5b), we model the bias as a quadratic polynomial.

**User-defined tolerance and baselines** We refer to the testing function proposed in this paper as $\hat{\phi}^{\text{CATE}}$, and we instantiate it using constant upper and lower bounds for the tolerance function: $\tau^{\text{os}}_\pm(X) = \tau^{\text{os}}(X) \pm \delta$ for some constant $\delta \in \mathbb{R}^+$. We compare our test against $\hat{\phi}^{\text{ATE}}$, which is a slight modification[1] of the test with tolerance proposed in (De Bartolomeis et al., 2024).

---

[1] $\hat{\phi}^{\text{ATE}}$ is a t-test for the null hypothesis that average treatment effects between the studies differ at most $\delta$.

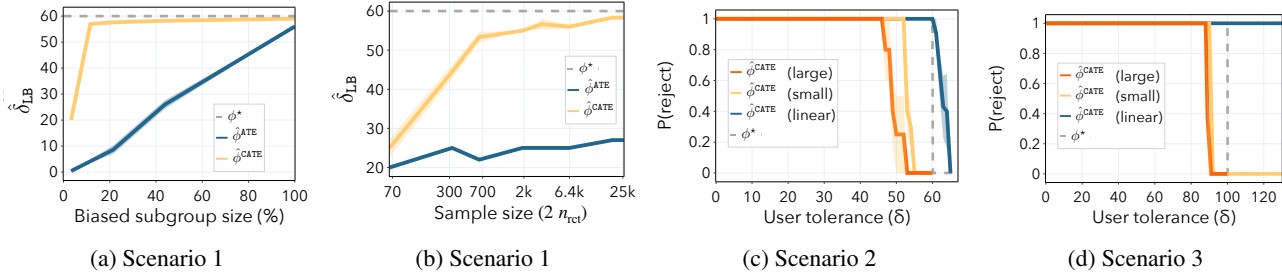

(a) Scenario 1          (b) Scenario 1          (c) Scenario 2          (d) Scenario 3

*Figure 2.* For all the plots: the significance level is set at $\alpha = 0.05$, $\phi^\star$ denotes the oracle test, which rejects for $\delta < \delta^\star$. (a-b) Scenario 1, comprising a single subgroup with a constant bias $\delta^\star = 60$: we plot the bias lower bound $\hat{\delta}_{\mathrm{LB}}$ as a function of (a) the biased subgroup percentage w.r.t. total sample size and (b) the randomized trial sample size. (c-d) Probability of rejection for different function classes $\mathcal{G}$ as a function of the user-specified tolerance $\delta$ for (c) Scenario 2 (Figure 5a) based on 12 subgroups with different biases and (d) Scenario 3 (Figure 5b) based on a quadratic polynomial bias. We report mean and standard error over 5 runs.

### 4.2. Experimental results

We first study the effect of the biased subgroup size (Figure 2a) and the randomized trial sample size (Figure 2b) on the lower bounds $\hat{\delta}_{\mathrm{LB}}$ obtained from our test $\hat{\phi}^{\mathrm{CATE}}$ and the baseline $\hat{\phi}^{\mathrm{ATE}}$. Next, we assess the validity and power of our test $\hat{\phi}^{\mathrm{CATE}}$ in two more complex settings: Scenario 2 (Figure 2c) and Scenario 3 (Figure 2d).

**Effect of biased subgroup and rct sample sizes** Figure 2a shows that our test yields an average lower bound $\hat{\delta}_{\mathrm{LB}}$ smaller and close to the true maximum bias $\delta^\star$. This implies that the test remains valid and exhibits significant power, even when the biased subgroup represents roughly 14% of the observational dataset. In contrast, $\hat{\phi}^{\mathrm{ATE}}$ experiences a significant drop in power as the proportion of biased data points decreases. Such behavior is expected since $\hat{\phi}^{\mathrm{ATE}}$ only tests for the difference of averages, and it cannot detect bias in small subgroups, i.e. it is not granular. In Figure 2b, we add a constant bias of 60 to 44% of the observational data points and study the effect of the randomized trial sample size. While our test suffers more than $\hat{\phi}^{\mathrm{ATE}}$ from a decrease in the sample size due to the use of kernels, it always yields higher power, even in the very small sample size regime with 70 data points.

**Validity and power in complex scenarios** Figure 2c and Figure 2d show the validity and power of our testing procedure for Scenario 2 (Figure 5a) and Scenario 3 (Figure 5b), respectively. In both scenarios, if we use a neural network to approximate the bias function, our test remains valid and shows very high power since it rejects the null hypothesis at values of $\delta$ close to the true bias $\delta^\star$.

**Effect of misspecified function class** Notably, when $g$ is modeled with a linear function, our test loses its validity, rejecting values of $\delta$ that are larger than the true bias. Such behavior is expected as the chosen function class $\mathcal{G}$ lacks the complexity necessary to capture the true bias model. Nevertheless, we observe that the *small* network with one hidden layer is already sufficient. Further, significantly increasing the complexity – the *large* network has approximately 45 times more parameters than the *small* one – still yields high power. Therefore, we recommend practitioners to be conservative in their choice of function class to ensure validity, even if it might come at the potential cost of some power. Although we cannot guarantee convergence to a global optimum, given the non-convexity of the problem for complex function classes, we show that the optimization procedure is stable and reaches the same solution in Appendix B.2.

## 5. Real-world experiments

In this section, we provide a concrete application of the benchmarking framework using the Women's Health Initiative (WHI) study. We show how tolerance and granularity are necessary for effective benchmarking.

### 5.1. The WHI controversy

The WHI study included a randomized trial and an observational study that investigated the use of hormone therapy (HT) for preventing common sources of mortality among postmenopausal women, including cardiovascular disease, cancer, and fractures (Anderson et al., 2003).

**To HT, or not to HT** The initial results of the WHI study in 2002 led to fear and confusion regarding the use of hormone therapy (HT) after menopause, resulting in a dramatic reduction in prescriptions for HT around the world. Although in 2002, it was stated that HT increases the risk of coronary heart disease (CHD) for all women, subsequent studies clearly showed that younger women close to menopause can benefit from HT. Further, subsequent randomized trials have continued demonstrating the benefits of HT when started early in young women close to

*Table 1.* The significance level is set at $\alpha = 0.05$. $\hat{\delta}_{\mathrm{CT}}$ is the amount of bias that would explain away the positive effect of HT in young women close to menopause. $\hat{\delta}_{\mathrm{LB}}$ is the maximum bias detected in the observational study. $\hat{\phi}^{\mathrm{ATE}}_{\delta=0}$ and $\hat{\phi}^{\mathrm{CATE}}_{\delta=0}$ denote the respective tests without tolerance, i.e. when the tolerance function is set at $\delta = 0$.

| Statistical tests | $\hat{\phi}^{\mathrm{CATE}}$ | $\hat{\phi}^{\mathrm{ATE}}$ | $\hat{\phi}^{\mathrm{CATE}}_{\delta=0}$ | $\hat{\phi}^{\mathrm{ATE}}_{\delta=0}$ |
|---|---|---|---|---|
| $\hat{\delta}_{\mathrm{CT}}$ | 0.32 | 0.32 | 0.32 | 0.32 |
| $\hat{\delta}_{\mathrm{LB}}$ | **0.25** | 0.11 | ✗ | ✗ |
| Reject the study | 0 | 0 | 1 | 1 |

menopause (Hodis et al., 2016; Taylor et al., 2017). To date, the consensus among epidemiologists is that hormone therapy reduces the risk of CHD in women aged less than 60 years and within 10 years of menopause; see e.g. the current guidelines for hormone therapy (Lee et al., 2020).

**Limitations of the WHI randomized trial**  The main issue with the randomized trial is that younger women's cardiac events are relatively rare. Indeed, not only would it have been prohibitively expensive to conduct a randomized trial exclusively in younger women, but it would have also taken many years to accumulate enough events to reach statistical significance. Hence, the trial lacked enough events to reach statistical significance on the subgroup of interest. On the other hand, the average treatment effect over all the patients suggested an increase in CHD risk because the majority of cardiac events came from older women, and epidemiologists concluded that HT is harmful to all women.

**Benchmarking can help!**  The natural question is, thus, if benchmarking the observational study could have prevented such a turn of events. Indeed, this is the perfect setting to test our methodology, as we would like to ask the question:

*Is the bias in the observational study enough to explain away the benefits of HT in young women close to menopause?*

In what follows, we show that answering such a question requires a statistical test that offers tolerance. Further, even though we cannot demonstrate that granularity is necessary in this concrete example[2], we stress that it is equally important in practice.

### 5.2. Experimental results

Linking back to our question of interest, we demonstrate how our method can provide a correct answer, i.e. one that

---

[2]To do so, we would need to know a small biased subgroup in the observational study and show that only the tests with granularity detect the bias. Unfortunately, we are unaware of subgroups that were found to be biased in the WHI study.

aligns with the epidemiology literature. A natural way to do so is to first estimate from the available data the amount of bias that would explain away the treatment effect on the group of interest, defined as

$$\hat{\delta}_{\mathrm{CT}} := \left| \mathbb{E}_{\mathbb{P}^{\mathrm{os}}} \left[ \tau^{\mathrm{os}}(X) \mid X \in G \right] \right|.$$

In essence, the critical value quantifies the minimum strength of bias for which positive and negative values of treatment effect are reasonable, thereby invalidating the observational study results[3]. In our example, the group $G$ is defined as young women (age $\leq 60$) who are close to menopause ($\leq 10$ years).

Similarly to the semi-synthetic experiments, we instantiate the tolerance functions using constant upper and lower bounds, i.e. $\tau^{\mathrm{os}}_{\pm}(X) = \tau^{\mathrm{os}}(X) \pm \delta$ for some constant $\delta \in \mathbb{R}^+$. We compute the lower bound $\hat{\delta}_{\mathrm{LB}}$ on the maximum amount of treatment effect bias in the observational study, as defined in Equation (5). We remark that this quantity can be computed only for tests that allow some tolerance. Then, our decision-making procedure will flag the observational study as invalid if $\hat{\delta}_{\mathrm{LB}} \geq \hat{\delta}_{\mathrm{CT}}$.

**Experimental details**  We consider a binary-valued outcome: the presence of coronary heart disease within the follow-up period. We choose as covariates $X$ the basic adjustment variables used in many existing analyses, and we further limit patients to those who were not current users of HT at the time of enrolment, as the duration of HT use has been found to have a substantial impact on treatment effects (Prentice et al., 2005; Vandenbroucke, 2009). We refer to Appendix C.2 for complete experimental details.

We present evidence that our procedure can yield the conclusions established in the epidemiological literature. It avoids issuing false alarms when the bias is negligible (tolerance) and detects a larger amount of bias, as it is more powerful than tests based on average treatment effect (granularity).

**Results**  In Table 1, we show the result for all the statistical tests on the WHI study. First, we observe that both tests that allow for tolerance correctly do not flag the study, while $\hat{\phi}^{\mathrm{CATE}}_{\delta=0}$ and $\hat{\phi}^{\mathrm{ATE}}_{\delta=0}$ do. This difference shows the importance of tolerance for distinguishing between small and large amounts of bias. Second, we observe that the lower bound on the bias is larger for the test with granularity $\hat{\phi}^{\mathrm{CATE}}$. Such behavior is expected and shows the importance of granularity to detect bias that would otherwise go unnoticed using the test without any granularity $\hat{\phi}^{\mathrm{ATE}}$.

---

[3]Note that other choices for the critical value are possible, and practitioners should determine the most appropriate one given the specific context.

## Acknowledgements

PDB was supported by the Hasler Foundation grant number 21050. JA was supported by the ETH AI Center. KD was supported by the ETH AI Center and the ETH Foundations of Data Science.

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

# Appendices

The following appendices provide deferred proofs, experiment details, and ablation studies.

## A. Methodology

For the sake of clarity, we write $n := n_{\mathrm{rct}}/2$, $\mathbb{P} := \mathbb{P}^{\mathrm{rct}}$ and $\mathbb{E}\left[\cdot\right] := \mathbb{E}_{\mathbb{P}^{\mathrm{rct}}}\left[\cdot\right]$ throughout this section.

### A.1. Proof of Theorem 3.1

We begin with the simple observation that

$$\min_{g \in \mathcal{G}} \left| \frac{\sqrt{n}\hat{\mathbb{H}}^2(\hat{\psi}_g)}{\hat{\sigma}\left(\hat{\mathbb{H}}^2(\hat{\psi}_g)\right)} \right| \leq \left| \frac{\sqrt{n}\hat{\mathbb{H}}^2(\hat{\psi}_{g^\star})}{\hat{\sigma}\left(\hat{\mathbb{H}}^2(\hat{\psi}_{g^\star})\right)} \right|,$$

which holds under the assumption that $g^\star \in \mathcal{G}$. Thus, asymptotic validity of $\hat{\phi}$ follows when showing that the RHS converges in distribution to an absolute normal distribution.

The key ingredient to prove this statement is to show that the following convergence in probability holds for all fixed $n$:

$$\hat{\mathbb{H}}^2(\hat{\psi}_{g^\star}) \to \hat{\mathbb{H}}^2(\psi_{g^\star}) \quad \text{and} \quad \hat{\sigma}^2\left(\hat{\mathbb{H}}^2(\hat{\psi}_{g^\star})\right) \to \hat{\sigma}^2\left(\hat{\mathbb{H}}^2(\psi_{g^\star})\right), \quad \text{as } n_{\mathrm{os}} \to \infty. \tag{6}$$

Then, under Assumption ($i$), we can apply Theorem 4.2 from Kim & Ramdas (2024) to show that

$$\frac{\sqrt{n}\hat{\mathbb{H}}^2(\psi_{g^\star})}{\hat{\sigma}\left(\hat{\mathbb{H}}^2(\psi_{g^\star})\right)} \to \mathcal{N}(0,1), \quad \text{as } n \to \infty.$$

Moreover, as a consequence of Equation (12) and (57) in the proof of Theorem 4.2 from Kim & Ramdas (2024), we have that

$$\frac{1}{n\hat{\sigma}^2\left(\hat{\mathbb{H}}^2(\psi_{g^\star})\right)} = O_{\mathbb{P}}(1).$$

Thus, when applying Slutsky's Theorem we have that

$$\frac{\sqrt{n}\hat{\mathbb{H}}^2(\hat{\psi}_{g^\star})}{\hat{\sigma}\left(\hat{\mathbb{H}}^2(\hat{\psi}_{g^\star})\right)} \to \mathcal{N}(0,1), \quad \text{as } n \to \infty \text{ and } n_{\mathrm{os}} \to \infty,$$

and the statement in Theorem 3.1 follows. It now remains to prove Equation 6.

PROOF OF STATEMENT IN EQUATION 6

We begin by defining the error term

$$\triangle := \hat{\psi}_{g^\star}(Z) - \psi_{g^\star}(Z) = g^\star(X^{\mathcal{J}})\left(\hat{\tau}_+^{\mathrm{os}}(X) - \tau_+^{\mathrm{os}}(X)\right) + (1 - g^\star(X^{\mathcal{J}}))\left(\hat{\tau}_-^{\mathrm{os}}(X) - \tau_-^{\mathrm{os}}(X)\right),$$

and we denote with $\triangle_i$ the i.i.d. samples from $\mathbb{P}$. We restate the definition of the mean and variance terms here. Formally, we split the dataset $D_{\mathrm{rct}}$ equally into two folds, $\mathcal{I}_1$ and $\mathcal{I}_2$, of size $n$ and obtain

$$n\hat{\mathbb{H}}^2(\hat{\psi}_{g^\star}) = \frac{1}{\sqrt{n}}\sum_{i \in \mathcal{I}_1}\hat{\psi}_i\frac{1}{\sqrt{n}}\sum_{j \in \mathcal{I}_2}k(X_i^{\mathcal{J}}, X_j^{\mathcal{J}})\hat{\psi}_j, \tag{7}$$

$$n\hat{\sigma}^2\left(\hat{\mathbb{H}}^2(\hat{\psi}_{g^\star})\right) = \frac{1}{n}\sum_{i \in \mathcal{I}_1}\hat{\psi}_i^2\left(\frac{1}{\sqrt{n}}\sum_{j \in \mathcal{I}_2}k(X_i^{\mathcal{J}}, X_j^{\mathcal{J}})\hat{\psi}_j\right)^2 - \left(\sqrt{n}\hat{\mathbb{H}}^2(\hat{\psi}_{g^\star})\right)^2, \tag{8}$$

where we use the shorthand $\hat{\psi}_i := \hat{\psi}_{g^\star}(Z_i)$ and $\psi_i := \psi_{g^\star}(Z_i)$.

**Preliminary step: bounds for the error term** $\triangle_i$    By Assumption $(ii)$ in Theorem 3.1, we have that

$$\mathbb{E}\left[\triangle^2\right] =: \|\triangle\|^2_{L_2(\mathbb{P})} \le 2\|\hat{\tau}^{\mathrm{os}}_+ - \tau^{\mathrm{os}}_+\|^2_{L_2(\mathbb{P})} + 2\|\hat{\tau}^{\mathrm{os}}_- - \tau^{\mathrm{os}}_-\|^2_{L_2(\mathbb{P})} = o_{\mathbb{P}^{\mathrm{os}}}\left(\frac{1}{n}\right), \tag{9}$$

where the probability $\mathbb{P}^{\mathrm{os}}$ is over the dataset $\mathcal{D}^{\mathrm{os}}$ used to train $\hat{\tau}^{\mathrm{os}}_{\pm}$. We further define

$$\tau_2(X^{\mathcal{J}}) := \frac{1}{\sqrt{n}}\sum_{j\in\mathcal{I}_2} k(X^{\mathcal{J}}_i, X^{\mathcal{J}}_j)\triangle_j \quad \text{and} \quad \tau_1(X^{\mathcal{J}}) := \frac{1}{\sqrt{n}}\sum_{i\in\mathcal{I}_1} k(X^{\mathcal{J}}_i, X^{\mathcal{J}}_j)\triangle_i.$$

We will repeatedly make use of the following bound, which holds analogously for $\tau_1$:

$$\sup_{X^{\mathcal{J}}} \left|\mathbb{E}\left[\tau_2(X^{\mathcal{J}})|X^{\mathcal{J}}\right]\right| = \sup_{X^{\mathcal{J}}} \left|\sqrt{n}\,\mathbb{E}\left[k(X^{\mathcal{J}}, \tilde{X}^{\mathcal{J}})\tilde{\triangle}|X^{\mathcal{J}}\right]\right| \lesssim \sqrt{n}\sqrt{\mathbb{E}\left[\triangle^2\right]},$$

where $\tilde{X}^{\mathcal{J}}$ and $\tilde{\triangle}$ are i.i.d. copies of $X^{\mathcal{J}}$ and $\triangle$, and in the last inequality, we used Cauchy-Schwartz together with the fact that the kernel is uniformly bounded. We will further use that

$$\sup_{X^{\mathcal{J}}}\left[\mathbb{E}\left[\left(\tau_2(X^{\mathcal{J}})\right)^2|X^{\mathcal{J}}\right]\right] = \sup_{X^{\mathcal{J}}}\left[\mathbb{E}\left[\frac{1}{n}\sum_{j\in\mathcal{I}_2}k(X^{\mathcal{J}}, X^{\mathcal{J}}_j)^2\triangle^2_j + \frac{1}{n}\sum_{\substack{j,j'\in\mathcal{I}_2\\ j\ne j'}}k(X^{\mathcal{J}}, X^{\mathcal{J}}_j)k(X^{\mathcal{J}}, X^{\mathcal{J}}_{j'})\triangle_j\triangle_{j'}\,\middle|\,X^{\mathcal{J}}\right]\right]$$

$$\lesssim \mathbb{E}\left[\triangle^2\right] + \sup_{X^{\mathcal{J}}}\left[\frac{n(n-1)}{n}\left(\mathbb{E}\left[k(X^{\mathcal{J}}, \tilde{X}^{\mathcal{J}})\tilde{\triangle}|X^{\mathcal{J}}\right]\right)^2\right]$$

$$\le \mathbb{E}\left[\triangle^2\right] + (n-1)\sup_{X^{\mathcal{J}}}\left[\mathbb{E}\left[k^2(X^{\mathcal{J}}, \tilde{X}^{\mathcal{J}})|X^{\mathcal{J}}\right]\mathbb{E}\left[\triangle^2\right]\right] = o_{\mathbb{P}^{\mathrm{os}}}(1),$$

where we used Cauchy-Schwartz again. We are now ready to show the convergences in Equation (6).

**Term 1: controlling $\hat{\mathbb{H}}^2(\hat{\psi}_{g^\star})$**    We first control the mean term $\hat{\mathbb{H}}^2(\hat{\psi}_{g^\star})$. Since $n$ is held fixed, it is equivalent to show that

$$\left|n\hat{\mathbb{H}}^2(\hat{\psi}_{g^\star}) - n\hat{\mathbb{H}}^2(\psi_{g^\star})\right| = o_{\mathbb{P}^{\mathrm{os}}}(1). \tag{10}$$

We decompose the difference into the following three terms:

$$n\hat{\mathbb{H}}^2(\hat{\psi}_{g^\star}) - n\hat{\mathbb{H}}^2(\psi_{g^\star}) = \underbrace{\frac{1}{\sqrt{n}}\sum_{i\in\mathcal{I}_1}\psi_i\tau_2(X^{\mathcal{J}}_i)}_{=:T_1} + \underbrace{\frac{1}{\sqrt{n}}\sum_{j\in\mathcal{I}_2}\psi_j\tau_1(X^{\mathcal{J}}_j)}_{=:T_2} + \underbrace{\frac{1}{\sqrt{n}}\sum_{i\in\mathcal{I}_1}\triangle_i\tau_2(X^{\mathcal{J}}_i)}_{=:T_3}. \tag{11}$$

To control the first two terms, we note that under the null hypothesis in Equation (2), it holds that $\mathbb{E}\left[\psi_{g^\star} \mid X^{\mathcal{J}} = x\right] = 0$, for all $x \in \mathrm{supp}\left(\mathbb{P}^{\mathrm{rct}}_{X^{\mathcal{J}}}\right)$. Thus, it suffices to show that the variance goes to zero:

$$\mathrm{Var}\left[\frac{1}{\sqrt{n}}\sum_{i\in\mathcal{I}_1}\psi_i\tau_2(X^{\mathcal{J}}_i)\right] = \mathbb{E}\left[\left(\frac{1}{\sqrt{n}}\sum_{i\in\mathcal{I}_1}\psi_i\tau_2(X^{\mathcal{J}}_i)\right)^2\right] = \mathbb{E}\left[\mathbb{E}\left[\psi^2|X^{\mathcal{J}}\right]\left(\tau_2(X^{\mathcal{J}})\right)^2\right]$$

$$\lesssim \mathbb{E}\left[\left(\tau_2(X^{\mathcal{J}})\right)^2\right] = o_{\mathbb{P}^{\mathrm{os}}}(1),$$

where we used that the conditional second moment $\mathbb{E}\left[\psi^2_{g^\star}|X^{\mathcal{J}}\right]$ is uniformly bounded, since the outcome $Y$ and the tolerance function $\tau^{\mathrm{os}}_{\pm}$ are both bounded. Further, the same argument also applies when swapping $\mathcal{I}_1$ with $\mathcal{I}_2$, we thus can conclude from Chebyshev's inequality that

$$|T_1| = o_{\mathbb{P}^{\mathrm{os}}}(1) \quad \text{and} \quad |T_2| = o_{\mathbb{P}^{\mathrm{os}}}(1).$$

Next, we bound the last term $T_3$. We first consider the mean of $T_3$:

$$\mathbb{E}\left[T_3\right] = \sqrt{n}\,\mathbb{E}\left[\triangle\tau_2(X^{\mathcal{J}})\right] = \sqrt{n}\,\mathbb{E}\left[\triangle\,\mathbb{E}\left[\tau_2(X^{\mathcal{J}})|X^{\mathcal{J}}\right]\right] \leq \sup_{X^{\mathcal{J}}}\left[\left|\mathbb{E}\left[\tau_2(X^{\mathcal{J}})|X^{\mathcal{J}}\right]\right|\right]\,\mathbb{E}\left[\sqrt{n}\,|\triangle|\right]$$

$$\leq \sup_{X^{\mathcal{J}}}\left[\left|\mathbb{E}\left[\tau_2(X^{\mathcal{J}})|X^{\mathcal{J}}\right]\right|\right]\sqrt{n}\sqrt{\mathbb{E}\left[\triangle^2\right]}$$

$$= o_{\mathbb{P}^{\text{os}}}(1).$$

Then, we consider the variance of $T_3$:

$$\mathbb{E}\left[T_3^2\right] = \mathbb{E}\left[\triangle^2\left(\tau_2(X^{\mathcal{J}})\right)^2\right] = \mathbb{E}\left[\triangle^2\,\mathbb{E}\left[\left(\tau_2(X^{\mathcal{J}})\right)^2|X^{\mathcal{J}}\right]\right] \leq \sup_{X^{\mathcal{J}}}\left[\mathbb{E}\left[\left(\tau_2(X^{\mathcal{J}})\right)^2|X^{\mathcal{J}}\right]\right]\mathbb{E}\left[\triangle^2\right] = o_{\mathbb{P}^{\text{os}}}(1).$$

Thus, we can conclude that $|T_3| = o_{\mathbb{P}}(1)$, and the equality in Equation (10) follows.

**Term 2: controlling** $\hat{\sigma}^2\left(\hat{\mathbb{H}}^2(\hat{\psi}_{g^\star})\right)$    As a second step, we control the variance term $\hat{\sigma}^2\left(\hat{\mathbb{H}}^2(\hat{\psi}_{g^\star})\right)$. Our goal is again to show that

$$\left|n\hat{\sigma}^2\left(\hat{\mathbb{H}}^2(\hat{\psi}_{g^\star})\right) - n\hat{\sigma}^2\left(\hat{\mathbb{H}}^2(\psi_{g^\star})\right)\right| = o_{\mathbb{P}^{\text{os}}}(1).$$

Given the results from the previous paragraph in Equation (10), we note that it suffices to show that

$$\left|\frac{1}{n}\sum_{i\in\mathcal{I}_1}\hat{\psi}_i^2\left(\frac{1}{\sqrt{n}}\sum_{j\in\mathcal{I}_2}k(X_i^{\mathcal{J}},X_j^{\mathcal{J}})\hat{\psi}_j\right)^2 - \frac{1}{n}\sum_{i\in\mathcal{I}_1}\psi_i^2\left(\frac{1}{\sqrt{n}}\sum_{j\in\mathcal{I}_2}k(X_i^{\mathcal{J}},X_j^{\mathcal{J}})\psi_j\right)^2\right| = o_{\mathbb{P}^{\text{os}}}(1). \tag{12}$$

We begin again by decomposing the difference of the two terms on the LHS into the following six terms:

$$= \underbrace{\frac{1}{n}\sum_{i\in\mathcal{I}_1}\triangle_i^2\left(\frac{1}{\sqrt{n}}\sum_{j\in\mathcal{I}_2}k(X_i^{\mathcal{J}},X_j^{\mathcal{J}})(\psi_j+\triangle_j)\right)^2}_{=:T_1} + \underbrace{\frac{1}{n}\sum_{i\in\mathcal{I}_1}(\psi_i+\triangle_i)^2\left(\tau_2(X_i^{\mathcal{J}})\right)^2}_{=:T_2} - \underbrace{\frac{1}{n}\sum_{i\in\mathcal{I}_1}\triangle_i^2\left(\tau_2(X_i^{\mathcal{J}})\right)^2}_{=:T_3}$$

$$+ \underbrace{\frac{2}{n}\sum_{i\in\mathcal{I}_1}\psi_i^2\left(\frac{1}{\sqrt{n}}\sum_{j\in\mathcal{I}_2}k(X_i^{\mathcal{J}},X_j^{\mathcal{J}})\psi_j\right)\tau_2(X_i^{\mathcal{J}})}_{=:T_4} + \underbrace{\frac{4}{n}\sum_{i\in\mathcal{I}_1}\psi_i\triangle_i\left(\frac{1}{\sqrt{n}}\sum_{j\in\mathcal{I}_2}k(X_i^{\mathcal{J}},X_j^{\mathcal{J}})\psi_j\right)\tau_2(X_i^{\mathcal{J}})}_{=:T_5}$$

$$+ \underbrace{\frac{2}{n}\sum_{i\in\mathcal{I}_1}\psi_i\triangle_i\left(\frac{1}{\sqrt{n}}\sum_{j\in\mathcal{I}_2}k(X_i^{\mathcal{J}},X_j^{\mathcal{J}})\psi_j\right)^2}_{=:T_6}.$$

We now show that $\forall i \in [1,\cdots,6], |T_i| = o_{\mathbb{P}^{\text{os}}}(1)$.

*Controlling $T_1$:* Since the term is non-negative, it suffices to show that the expectation $\mathbb{E}\left[T_1\right] = o_{\mathbb{P}^{\text{os}}}(1)$ and then apply

Markov's inequality. More formally, we have

$$
\mathbb{E}\left[T_1\right] = \mathbb{E}\left[\triangle^2 \left(\frac{1}{\sqrt{n}} \sum_{j \in \mathcal{I}_2} k(X^{\mathcal{J}}, X_j^{\mathcal{J}})(\psi_j + \triangle_j)\right)^2\right]
$$

$$
= \mathbb{E}\left[\triangle^2 \left[\frac{1}{n} \sum_{j \in \mathcal{I}_2} k^2(X^{\mathcal{J}}, X_j^{\mathcal{J}})\psi_j^2 + \left(\frac{1}{\sqrt{n}} \sum_{j \in \mathcal{I}_2} k(X^{\mathcal{J}}, X_j^{\mathcal{J}})\triangle_j\right)^2 + \frac{1}{n} \sum_{j \in \mathcal{I}_2} k^2(X^{\mathcal{J}}, X_j^{\mathcal{J}})\psi_j\triangle_j\right]\right]
$$

$$
= \mathbb{E}\left[\triangle^2 k(X^{\mathcal{J}}, \tilde{X}^{\mathcal{J}})^2[\tilde{\psi}^2 + \tilde{\psi}\tilde{\triangle}]\right] + \mathbb{E}\left[\triangle^2 \left(\tau_2(X^{\mathcal{J}})\right)^2\right]
$$

$$
\lesssim \mathbb{E}\left[\triangle^2\right]\left[\mathbb{E}\left[\psi^2\right] + \sqrt{\mathbb{E}\left[\psi^2\right]\mathbb{E}\left[\triangle^2\right]} + \sup_{X^{\mathcal{J}}} \mathbb{E}\left[\left(\tau_2(X^{\mathcal{J}})\right)^2 | X^{\mathcal{J}}\right]\right]
$$

$$
= o_{\mathbb{P}^{\mathrm{os}}}(1),
$$

where in the second equality we use again $\mathbb{E}\left[\psi_{g^\star} | X^{\mathcal{J}} = x\right] = 0$, for all $x \in \operatorname{supp}\left(\mathbb{P}_{X^{\mathcal{J}}}^{\mathrm{rct}}\right)$.

*Controlling $T_2$ and $T_3$:* We can again upper-bound the expectation and apply Markov's inequality. We have

$$
\mathbb{E}\left[T_2\right] \leq \mathbb{E}\left[\left(2\psi^2 + 2\triangle^2\right)\mathbb{E}\left[\left(\tau_2(X^{\mathcal{J}})\right)^2 | X^{\mathcal{J}}\right]\right] = \sup_{X^{\mathcal{J}}}\left[\mathbb{E}\left[\left(\tau_2(X^{\mathcal{J}})\right)^2 | X^{\mathcal{J}}\right]\right]\left(2\mathbb{E}\left[\psi^2\right] + 2\mathbb{E}\left[\triangle^2\right]\right) = o_{\mathbb{P}^{\mathrm{os}}}(1),
$$

and thus it also follows that $\mathbb{E}\left[T_3\right] = o_{\mathbb{P}^{\mathrm{os}}}(1)$.

*Controlling $T_4$, $T_5$ and $T_6$:* We note that the expectations $\mathbb{E}\left[T_4\right] = 0$, $\mathbb{E}\left[T_5\right] = 0$ and $\mathbb{E}\left[T_6\right] = 0$ are all zero. Thus, we can bound the terms in probability by showing that the respective variances converge to zero and then applying Chebyshev's inequality. We first upper-bound the variance of $T_4$:

$$
\operatorname{Var}\left[T_4\right] = \operatorname{Var}\left[\frac{2}{n} \sum_{i \in \mathcal{I}_1} \psi_i^2 \left(\frac{1}{\sqrt{n}} \sum_{j \in \mathcal{I}_2} k(X_i^{\mathcal{J}}, X_j^{\mathcal{J}})\psi_j\right) \tau_2(X_i^{\mathcal{J}})\right]
$$

$$
= \frac{4}{n}\mathbb{E}\left[\psi^4 \left(\frac{1}{\sqrt{n}} \sum_{j \in \mathcal{I}_2} k(X^{\mathcal{J}}, X_j^{\mathcal{J}})\psi_j\right)^2 \left(\tau_2(X^{\mathcal{J}})\right)^2\right]
$$

$$
= 4\mathbb{E}\left[\psi^4 \left(\frac{1}{n} \sum_{j \in \mathcal{I}_2} k(X^{\mathcal{J}}, X_j^{\mathcal{J}})\psi_j\right)^2 \left(\tau_2(X^{\mathcal{J}})\right)^2\right]
$$

$$
\lesssim \sup_{X^{\mathcal{J}}} \mathbb{E}\left[\left(\tau_2(X^{\mathcal{J}})\right)^2 | X^{\mathcal{J}}\right]
$$

$$
= o_{\mathbb{P}^{\mathrm{os}}}(1),
$$

where we use the fact that both the kernel $k$ and the fourth conditional moment of $\psi_{g^\star} | X^{\mathcal{J}}$ are almost surely upper bounded by a constant. Next, we bound the variance of $T_5$:

$$
\operatorname{Var}\left[T_5\right] = \operatorname{Var}\left[\frac{2}{n} \sum_{i \in \mathcal{I}_1} \psi_i\triangle_i \left(\frac{1}{\sqrt{n}} \sum_{j \in \mathcal{I}_2} k(X_i^{\mathcal{J}}, X_j^{\mathcal{J}})\psi_j\right) \tau_2(X_i^{\mathcal{J}})\right]
$$

$$
= 4\mathbb{E}\left[\psi^2\triangle^2 \left(\frac{1}{n} \sum_{j \in \mathcal{I}_2} k(X^{\mathcal{J}}, X_j^{\mathcal{J}})\psi_j\right)^2 \left(\tau_2(X^{\mathcal{J}})\right)^2\right]
$$

$$
= o_{\mathbb{P}^{\mathrm{os}}}(1).
$$

Finally, we upper-bound the variance of the term $T_6$:

$$
\begin{aligned}
\mathrm{Var}\left[T_6\right] &= \mathrm{Var}\left[\frac{2}{n}\sum_{i\in\mathcal{I}_1}\psi_i\triangle_i\left(\frac{1}{\sqrt{n}}\sum_{j\in\mathcal{I}_2}k(X_i^{\mathcal{J}},X_j^{\mathcal{J}})\psi_j\right)^2\right] \\
&= \frac{4}{n}\mathbb{E}\left[\psi^2\triangle^2\left(\frac{1}{n^2}\sum_{j\in\mathcal{I}_2}k(X^{\mathcal{J}},X_j^{\mathcal{J}})^4\psi_j^4 + \frac{6}{n^2}\sum_{j,j'\in\mathcal{I}_2;j\neq j'}k(X^{\mathcal{J}},X_j^{\mathcal{J}})^2k(X^{\mathcal{J}},X_{j'}^{\mathcal{J}})^2\psi_j^2\psi_{j'}^2\right)\right] \\
&= o_{\mathbb{P}^{\mathrm{os}}}(1).
\end{aligned}
$$

As a result, we conclude that $|T_4| = o_{\mathbb{P}^{\mathrm{os}}}(1)$, $|T_5| = o_{\mathbb{P}^{\mathrm{os}}}(1)$, and $|T_6| = o_{\mathbb{P}^{\mathrm{os}}}(1)$.

**Discussion of assumptions** Assumption (*i*) is mild and applies to very general settings, e.g. it is satisfied when $Y$ is a non-deterministic random variable. Assumption (*ii*) is stronger and generally only expected to hold when $n_{\mathrm{os}} \gg n_{\mathrm{rct}}$ and the support of the randomized control trial is contained in the support of the observational study, i.e. $\mathrm{supp}(\mathbb{P}_X^{\mathrm{rct}}) \subseteq \mathrm{supp}(\mathbb{P}_X^{\mathrm{os}})$. These two conditions are realistic in our setting, as they align with the standard design of observational studies (Franklin et al., 2019; Schurman, 2019; He et al., 2020). Further, we remark that previous works either assume oracle access to the functions $\tau_\pm^{\mathrm{os}}$ (Hussain et al., 2023; Demirel et al., 2024) or impose similar assumptions on the rates (De Bartolomeis et al., 2024).

**Why not a classic U-statistic?** We remark that it is not clear how to test the null hypothesis $H_0^{\mathcal{G}}$ using the classic U-statistic (Serfling, 1980), as done in previous works (see e.g. (Hussain et al., 2023; Demirel et al., 2024)). The main challenge is that under the null hypothesis $\mathbb{H}^2(\psi_{g^\star}) = 0$, the U-statistic converges in distribution to a weighted $\chi^2$-statistic. However, estimating the quantiles (needed for a valid test) of this asymptotic distribution via bootstrapping requires knowing the function $g^\star$ (Huskova & Janssen, 1993). In contrast, our test statistic $\mathbb{H}_{\mathrm{OPT}}^2$ is bounded by a valid asymptotic pivot, i.e. a function of the data and the unknown function $g^\star$ whose asymptotic distribution does not depend on $g^\star$. Hence, we can compute the quantiles of the RHS in Equation (4) and construct an asymptotically valid test.

## A.2. Power of the test

We first discuss a simple result on the power of the test from Equation (4) in rejecting an alternative hypothesis. While Hussain et al. (2023); Muandet et al. (2020) show asymptotic normality for the kernel conditional moment test statistics $\mathbb{M}^2$ under the alternative hypothesis that $\mathbb{M}^2 \neq 0$, the same result does not directly apply to our test statistic. However, as we show in the following theorem, our test statistic grows at a rate $\sqrt{n}$. For the sake of clarity, we only prove the result for the oracle test statistic $\mathbb{H}_{\mathrm{OPT}}^2$ computed from $\psi_g$. Nevertheless, we remark that our result can be easily extended to the empirical test statistics via the same argument used in the proof of Theorem 3.1.

**Theorem A.1.** *Assume that for every $\epsilon > 0$, the function class $\mathcal{G}$ has a finite $\ell_\infty-$norm covering number. Then, we can lower-bound the test statistic, in probability as $n \to \infty$, by*

$$
\mathbb{H}_{OPT}^2 = \min_{g\in\mathcal{G}}\left|\frac{\sqrt{n}\,\hat{\mathbb{H}}^2(\psi_g)}{\hat{\sigma}\left(\hat{\mathbb{H}}^2(\psi_g)\right)}\right| \gtrsim \sqrt{n}\left(\inf_{g\in\mathcal{G}}\sup_{\|f\|_{\mathcal{F}}\leq 1}\mathbb{E}\left[\psi_g(Z)f(X^{\mathcal{J}})\right]\right)^2,
$$

*where we use $\gtrsim$ to hide universal constants not depending on $n$.*

Thus, under the alternative hypothesis

$$
H_A^{\mathcal{G}}:\inf_{g\in\mathcal{G}}\sup_{\|f\|_{\mathcal{F}}\leq 1}\mathbb{E}_{\mathbb{P}^{\mathrm{rct}}}\left[\psi_g(Z)f(X^{\mathcal{J}})\right] > 0,
$$

the RHS grows at a rate $\sqrt{n}$, which implies that our test has an asymptotic power of one (note that the same rate is achieved by existing conditional moment tests (Hussain et al., 2023; Muandet et al., 2020)).

**Proof of Theorem A.1** Let us define

$$
T := \inf_{g\in\mathcal{G}}\sup_{\|f\|_{\mathcal{F}}\leq 1}\mathbb{E}\left[\psi_g(Z)f(V)\right],
$$

and note that if $T = 0$ the result follows trivially. Thus, we may assume that $T > 0$ is some constant independent of $n$. Additionally, observe that $\psi_g$ is uniformly bounded since the outcome $Y$ is a bounded random variable. Therefore, the variance term $\hat{\sigma}\left(\hat{\mathbb{H}}^2(\psi_g)\right)$ is also uniformly bounded, and it suffices to show that $\hat{\mathbb{H}}^2(\psi_g) = \Omega_{\mathbb{P}}(1)$ is lower bounded in probability.

**Controlling $\hat{\mathbb{H}}^2(\psi_g)$**   First, recall that our test statistic is given by

$$\hat{\mathbb{H}}^2(\psi_g) = \frac{1}{n^2} \sum_{i=1}^{n} \sum_{j=n+1}^{2n} \psi_g(Z_i) k(X_i^{\mathcal{J}}, X_j^{\mathcal{J}}) \psi_g(Z_j). \tag{13}$$

Further, for all $g \in \mathcal{G}$, it holds that

$$\mathbb{E}\left[\hat{\mathbb{H}}^2(\psi_g)\right] = \mathbb{E}\left[\psi_g(Z) k(X^{\mathcal{J}}, \tilde{X}^{\mathcal{J}}) \psi_g(\tilde{Z})\right] \geq \inf_{g \in \mathcal{G}} \mathbb{E}\left[\psi_g(Z) k(X^{\mathcal{J}}, \tilde{X}^{\mathcal{J}}) \psi_g(\tilde{Z})\right] = T^2,$$

where $\tilde{Z}$ is an independent copy of $Z$ following the same distribution, and the last equality follows from Equation (3). Thus, it suffices to show that the following inequality holds with probability one as $n \to \infty$

$$\sup_{g \in \mathcal{G}} \left|\hat{\mathbb{H}}^2(\psi_g) - \mathbb{E}\left[\hat{\mathbb{H}}^2(\psi_g)\right]\right| \leq \frac{T^2}{2}.$$

We use a simple $\epsilon$-net argument to show this result. Let $\mathcal{G}_\epsilon$ be the epsilon net in $\ell_\infty$ distance of balls with radii $\epsilon$. Then, since $\psi_g$ is uniformly bounded, it holds that for all $Z$ and $g \in \mathcal{G}$,

$$\inf_{\tilde{g} \in \mathcal{G}_\epsilon} |\psi_g(Z) - \psi_{\tilde{g}}(Z)| \lesssim \epsilon.$$

Thus, from the definition of $\hat{\mathbb{H}}^2(\psi_g)$ in Equation (13) it follows that we can choose a constant $\epsilon > 0$, such that the following inequality holds almost surely,

$$\sup_{g \in \mathcal{G}} \inf_{\tilde{g} \in \mathcal{G}_\epsilon} \left|\hat{\mathbb{H}}^2(\psi_g) - \hat{\mathbb{H}}^2(\psi_{\tilde{g}})\right| \leq \frac{T^2}{4}. \tag{14}$$

Then, for any constant $c > 0$, it holds that

$$\mathbb{P}\left(\sup_{g \in \mathcal{G}} \left|\hat{\mathbb{H}}^2(\psi_g) - \mathbb{E}\left[\hat{\mathbb{H}}^2(\psi_g)\right]\right| \leq \frac{T^2}{2}\right)$$

$$= \mathbb{P}\left(\sup_{g \in \mathcal{G}} \inf_{\tilde{g} \in \mathcal{G}_\epsilon} \left|\hat{\mathbb{H}}^2(\psi_g) - \hat{\mathbb{H}}^2(\psi_{\tilde{g}}) + \hat{\mathbb{H}}^2(\psi_{\tilde{g}}) - \mathbb{E}[\hat{\mathbb{H}}^2(\psi_{\tilde{g}})] + \mathbb{E}[\hat{\mathbb{H}}^2(\psi_{\tilde{g}})] - \mathbb{E}[\hat{\mathbb{H}}^2(\psi_g)]\right|\right)$$

$$\overset{(i)}{\geq} \mathbb{P}\left(\sup_{\tilde{g} \in \mathcal{G}_\epsilon} \left|\hat{\mathbb{H}}^2(\psi_{\tilde{g}}) - \mathbb{E}\left[\hat{\mathbb{H}}^2(\psi_{\tilde{g}})\right]\right| \leq 2c\right)$$

$$\overset{(ii)}{\geq} 1 - \sum_{\tilde{g} \in \mathcal{G}_\epsilon} \underbrace{\mathbb{P}\left(\left|\hat{\mathbb{H}}^2(\psi_{\tilde{g}}) - \mathbb{E}\left[\hat{\mathbb{H}}^2(\psi_{\tilde{g}})\right]\right| \geq 2c\right)}_{n \overset{\to}{\to} \infty \ 0 \ \text{(L.L.N.)}},$$

where (i) follows from applying the inequality in Equation (14) and (ii) follows since, by assumption, for every fixed $\epsilon > 0$ the cover $|\mathcal{G}_\epsilon| < \infty$ is constant as a function of $n$.

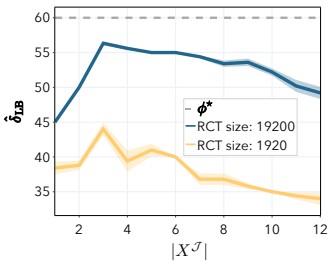
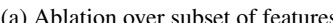

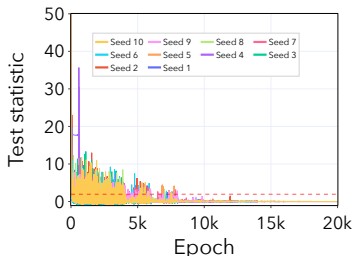

(a) Ablation over subset of features

(b) Optimization convergence

*Figure 3.* For all the plots: the significance level is set at $\alpha = 0.05$, and the bias model is from Scenario 2. (a) Effect of varying the feature set $X^{\mathcal{J}}$ on the average lower bound $\hat{\delta}_{\text{LB}}$, illustrating the trade-off between feature set size and the power of the test. $\phi^{\star}$ represents the oracle test, which rejects for $\delta < \delta^{\star}$. The highest power is achieved when the feature set size $|X^{\mathcal{J}}| = 3$, including only the relevant features to model the bias. We average runs over 5 seeds and report the standard error. (b) Evolution of the test statistic with respect to the training epochs using the small neural network. We set the user tolerance to $\delta = 58$, close to the maximum true bias $\delta^{\star} = 60$. The dashed red line represents the $\alpha$-quantile of the absolute normal distribution.

# B. Additional experiments

## B.1. Ablation study of the feature subset $X^{\mathcal{J}}$

In Scenario 2 from Figure 5a, we introduced constant bias in the subgroups resulting from different combinations of the features `newbie`, `mens` and `channel`, with a maximum true bias $\delta^{\star} = 60$. Figure 3a shows the effect of the selected feature set $X^{\mathcal{J}}$ on the average lower bound $\hat{\delta}_{\text{LB}}$ for the bias model from Scenario 2. When $|X^{\mathcal{J}}| = 3$, we select the features that capture the bias between rct and os datasets (`newbie`, `mens`, `channel`), and hence we achieve the highest power. Intuitively, if the feature set is smaller, some of the bias averages out, and the test loses power. On the other hand, when increasing the feature set, the test loses power due to the curse of dimensionality, being particularly severe with smaller sample sizes. After $|X^{\mathcal{J}}| = 6$ (i.e. $X^{\mathcal{J}} = X$), we add redundant features sampled from a standard normal distribution $\mathcal{N}(0, 1)$.

## B.2. Convergence of the optimization procedure

We provide evidence that our testing procedure is reliable, meaning that the optimizer consistently reaches the same solution for the bias model from Scenario 2 and the small neural network model. Recall that, given the non-convex nature of the optimization problem, we cannot guarantee convergence to the true global minimum $g^{\star}$. Figure 3b shows the test statistic as a function of the training epoch under different random network initializations. We observe that the test statistic consistently reaches the same minimum and that the optimization stabilizes after 10000 epochs.

## B.3. Interpretability of the testing procedure

Similar to the test proposed by Hussain et al. (2023), our testing procedure outputs a "witness function" that enables practitioners to identify the most biased subgroups within the observational dataset. Additionally, our witness function provides insights into the bias strength and direction for each subgroup. This is achieved by minimizing the objective in Equation (4), where we learn the bias function $\hat{g}$. If the function class $\mathcal{G}$ is sufficiently rich to model the bias structure, and the optimizer converges to the global minima, we expect $\hat{g}$ to be a good approximation of $g^{\star}$.

To interpret this bias function, we observe that $\hat{g}(X) \in [0, 1]$ interpolates between the tolerance bounds $\tau_{-}^{\text{os}}(X)$ and $\tau_{+}^{\text{os}}(X)$; therefore, values close to zero indicate a negative bias of magnitude close to user-tolerance $\delta$, while values close to one indicate the same for positive bias. Hence, we can estimate the subgroup bias as

$$\texttt{bias}(G) = \hat{\delta}_{\text{LB}} \left( \frac{2}{|G|} \sum_{X_i \in G} \hat{g}(X_i) - 1 \right), \tag{15}$$

where $G$ represents the subgroup of interest. In Figure 4, we illustrate how practitioners could use the witness function for Scenario 2, where the categorical nature of the features defines subgroups. We compare the estimated bias with the ground

truth and observe that our estimates closely align with the true bias model. In scenarios where subgroups are not predefined, a practitioner can select the bottom or top 10% of witness function values, as suggested by Hussain et al. (2023).

However, it is important to note that, unlike the approach by Hussain et al. (2023), we do not have guarantees for the correctness of the witness function, i.e. we cannot guarantee that $\hat{g} \to g^\star$. Therefore, any claims based on it should be approached cautiously and contrasted with domain expertise.

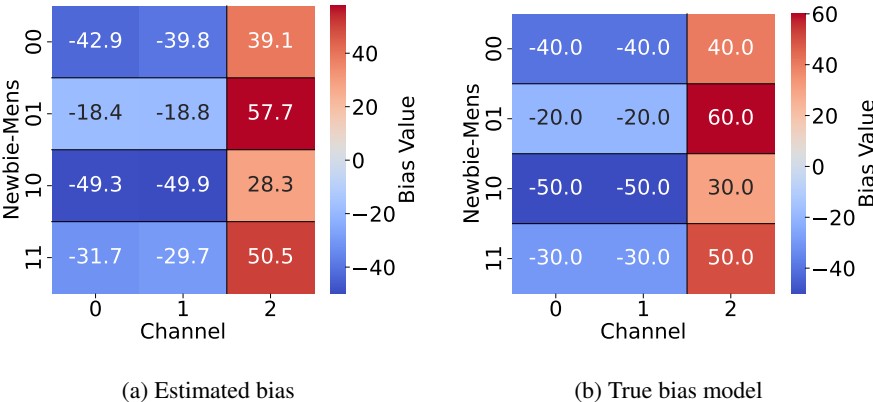

(a) Estimated bias                    (b) True bias model

*Figure 4.* Comparison between the estimated and true bias models for Scenario 2. Our estimates of the bias from Equation (15) closely align with the true bias. We run the test with a random seed, using the same hyperparameters as in our experimental evaluation, and set the user tolerance to $\delta = 57$.

# C. Experimental details

## C.1. Hillstrom's MineThatData

Hillstrom's MineThatData Email dataset (Hillstrom, 2008) is a large-scale, real-world randomized trial that contains records of 64,000 customers who made purchases online within the last twelve months. They were part of an email campaign designed to assess the effectiveness of different campaign strategies. Two treatment groups, "Men's" and "Women's" email campaigns, and a control group were established, with treatments randomly assigned. Our analysis primarily focuses on a combined treatment group, which constitutes approximately 66% of the dataset. Although the original dataset presents various outcomes, including binary indicators of customers visiting or purchasing in the days after the campaign, we focus on the dollars spent in the two weeks post-campaign. The dataset provides data on annual spending (`history`), merchandise type (`mens` and `womens`), geographical location (`zip code`), newcomer status (`newbie`), and purchasing avenues (`channel`). We, therefore, discard features describing the history segment (`history segment`) and recency of the last purchase (`recency`). Since the average treatment effect is close to zero, we add a constant shift of 30 to all treated individuals, allowing us more flexibility to introduce bias. We normalize continuous features and one-hot-encode categorical features, resulting in a 13-dimensional dataset. By default, we use 80% of the full dataset as the observational study (os), and the remaining 20% as the randomized controlled trial (rct).

We fit the propensity score using logistic regression with default hyperparameters from `scikit-learn`. We train a `Random Forest Classifier` for the selection score (rct or os), also with default hyperparameters from `scikit-learn`. Finally, we estimate the CATE functions using the doubly-robust learner from Kennedy (2023), instantiating `Random Forest Regressors` for the potential outcome functions and the pseudo-outcome regression, fixing hyperparameters to 300 `tree estimators` with a `maximum depth` of 6.

**Bias models** We illustrate the bias model for Scenario 2 and Scenario 3 in Figure 5. For scenario 3, we sample the coefficient for the polynomial bias model in Figure 5a from a normal distribution $\mathcal{N}(0, 0.01^2)$.

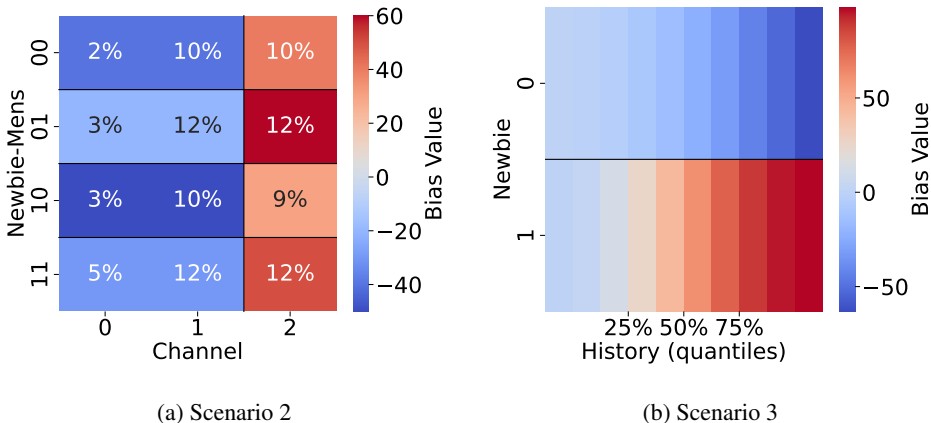

(a) Scenario 2                                      (b) Scenario 3

*Figure 5.* Heatmap visualizations of the bias for (a) Scenario 2 based on 12 subgroups with different biases (the numbers in the cells represent the percentage w.r.t. the full observational dataset), and (b) Scenario 3 based on a quadratic polynomial bias.

**Implementation** We use the Laplacian kernel with a scale of 1.0 to compute our test statistic $\hat{\phi}^{\text{CATE}}$. We perform gradient descent for 6000 epochs using the Adam optimizer from the JAX-based library `optax` with its default hyperparameters and record the smallest test statistic. As function class $\mathcal{G}$, we consider linear functions and two multilayer perceptrons (MLPs), one *small* and one *large*, with hidden layer widths of 10 and 100-50-10-5 neurons, respectively. For the linear function and the small MLP, we set the learning rate to 0.1, and for the large MLP, we set it to 0.01. For the test $\hat{\phi}^{\text{ATE}}$, we use 500 bootstrap samples to estimate the variance of the test statistic.

## C.2. Women's Health Initiative

The Women's Health Initiative (WHI) is a long-term national health study that has focused on strategies for preventing the major causes of death, disability, and frailty in older women, specifically heart disease, cancer, and osteoporotic fractures.

This multi-million dollar, 20+ year project, sponsored by the National Institutes of Health (NIH) and the National Heart, Lung, and Blood Institute (NHLBI), initially enrolled 161,808 women aged 50-79 between 1993 and 1998. The WHI was one of the most definitive, far-reaching clinical trials of post-menopausal women's health ever undertaken in the US.

The WHI had two major parts: a randomized trial and an observational study. The randomized trial enrolled 68,132 women in trials testing three prevention strategies. Eligible women could choose to enroll in one, two, or three of the trial components.

- A Hormone Therapy Trial (HT) that examined the effects of combined hormones or estrogen alone on the prevention of heart disease and osteoporotic fractures and associated risk for breast cancer.

- A Dietary Modification Trial (DM) that evaluated the effect of a low-fat and high-fruit, vegetable, and grain diet on preventing breast and colorectal cancers and heart disease.

- A Calcium and Vitamin D Trial (CaD) that evaluated the effect of calcium and vitamin D supplementation on preventing osteoporotic fractures and colorectal cancer.

The Observational Study (OS) examines the relationship between lifestyle, health risk factors, and disease outcomes. This component involves tracking the medical events and health habits of 93,676 women. Recruitment for the observational study was completed in 1998, and participants have been followed since.

We use observational study and randomized trial data from the Women's Health Initiative (WHI) to assess our method in a real-world scenario. We use the Hormone Therapy (HT) trial as the RCT in our analysis ($n_{\text{rct}} = 16,608$), run on postmenopausal women aged 50-79 years with an intact uterus. The trial investigated the effect of hormone therapy on several types of cancers, cardiovascular events, and fractures, measuring the "time-to-event" for each outcome. In the WHI setup, the observational study component was run in parallel, and outcomes were tracked similarly to those of the RCT.

**Data preprocessing**   We binarize a composite outcome, where $Y = 1$ if coronary heart disease was observed in the first seven years of follow-up, and $Y = 0$ otherwise. To establish treatment and control groups in the observational study, we use questionnaire data in which participants confirm or deny usage of combination hormones (i.e. both estrogen and progesterone) in the first three years. Using this procedure, we end up with a total of $n_{\text{os}} = 33,511$ patients. Finally, we restrict the set of covariates used to those that are measured in both the RCT and the observational study. In particular, we use as covariates only those measured in both the RCT and observational study, and we further restrict them to those identified as significant in epidemiological literature, such as in (Prentice et al., 2005). Specifically, the covariates in our analysis are: AGE, ETHNIC_White, BMI, SMOKING_Past_Smoker, SMOKING_Current_Smoker, EDUC_x_College_graduate_or_Baccalaureate Degree, EDUC_x_Some_post-graduate_or_professional, MENO, PHYSFUN. The data used is available on BIOLINCC.

**Experimental details**   We use a gaussian kernel with bandiwidth = 1.0. The set of features for the granularity of the test is chosen to be $J = \{\texttt{AGE}, \texttt{MENO}\}$. We use a logistic regression model for both the outcome model and propensity score (default hyperparameters in scikit-learn were used). We train a neural (1 hidden layer and 10 neurons) network with Adam, with a learning rate of 0.01 for 500 epochs. We repeat the optimization for 10 seeds with different initializations to ensure that we converge.