# OpenReview forum: "Detecting critical treatment effect bias in small subgroups"
_ICML.cc/2024/Workshop/ML4LMS — ML4LMS Poster_

### Official Review · Reviewer_5tqU · 2024-05-31

**Rating:** 10
**Confidence:** 3

**Review:**

## Summary:

The paper highlights the challenge of generalizability in treatment effect estimates derived from randomized controlled trials (RCTs) versus observational studies. It proposes a novel statistical method to detect and quantify treatment effect bias in small subgroups by comparing observational study estimates against benchmarks derived from RCTs. Key contributions include: i) a kernelized test statistic to identify discrepancies between treatment effect estimates from RCTs and observational studies. ii) theoretical analysis including a lower bound on the maximum bias strength. iii) application to a real-world randomized trial dataset.

## Strength:

The paper is well-written. All mathematical notations are well explained. In particular, the methodology section is thorough and well-structured.

The use of a signal function and a kernelized test statistic are key to addressing the problem of treatment effect bias in small subgroups. By addressing both tolerance and granularity, the method is particularly suited for detecting subtle biases that could significantly impact clinical decision-making.

## Comments on general applicability, choice of dataset, and potential direction of improvements:

There is a lack of discussion on the computational complexity and feasibility of the proposed test in large-scale studies.

The paper assumes the availability of a sufficiently rich function class \( G \) for the signal function \( \psi_g(Z) \). However, the practical aspects of selecting and validating such a function class are not discussed.

The results section could provide more details on the datasets used for validation. Details on the size, characteristics, and any preprocessing steps taken would improve the readability and reproducibility of the study.

The performance of the method under different scenarios or sensitivity to various types of bias is not thoroughly explored.
Usually participants do not adhere to their assigned treatment. For example, in a drug trial, some patients assigned to the treatment group might not take the medication as prescribed, or patients in the control group might obtain the medication outside the study. How reasonable is it to model such “non-compliance” assumptions?

Usually there exist confounders that are not measured or controlled for in the analysis. For example, a socioeconomic status could affect both the likelihood of receiving a treatment and the health outcome, failing to measure and adjust for that can bias the treatment effect estimate. Are there simulation based data where such real world situations are accounted for to observe the impact on bias estimates. Perhaps include robustness checks that test the stability of the results under different assumptions or when different subsets of data are used.

More discussion on the applicability of the method across different medical domains and types of treatment effects would be beneficial.

Line 40: states discrepancy between the two groups. Would be good to discuss the issues that lead to such discrepancy.

Overall, I find it is a very good paper proposing a novel statistical test for benchmarking observational studies against randomized trials. Supported by experiments on real world data.

---

### Official Review · Reviewer_LKfa · 2024-06-12
**Detecting bias in treatment effects of observational studies by benchmarking with randomised trials**

**Rating:** 8
**Confidence:** 2

**Review:**

# Summary:
This paper proposes a new technique to estimate bias in treatment effects from observational studies by comparison against the randomised trials. The main contribution of this work is a proposed statistical test for the null hypothesis, which compares the treatment effects from the observational studies and randomised trials.
# Strengths and Weaknesses:
The work's strengths include a strong theoretical framework to back the experimental results and real-life applications.
The strategy works well in the chosen example of the Women's Health Initiative(WHI) study; however, it would be interesting to see how this performs outside the medical domain, such as in socio-economic interventions and on more complex datasets and real-life examples.
# Presentation:
The writing is structured very well. The authors have provided detailed evaluations, comparisons and ablations. They have provided ample evidence and support through theorems and theoretical/mathematical proofs wherever required.
Although the other sections of the work cover all the previous works and approaches, it would have been nice to have dedicated limitations, future work, conclusion and impact sections at the end.
# Soundness:
4 (Excellent)
# Contribution:
3 (Good)